# The Role of NKT Cells in Glioblastoma

**DOI:** 10.3390/cells10071641

**Published:** 2021-06-30

**Authors:** Emily E. S. Brettschneider, Masaki Terabe

**Affiliations:** 1Neuro-Oncology Branch, Center for Cancer Research, National Cancer Institute, NIH, Bethesda, MD 20892, USA; emily.steffke@ludwig.ox.ac.uk; 2Nuffield Department of Medicine, Ludwig Institute for Cancer Research, University of Oxford, Oxford OX3 7DQ, UK

**Keywords:** glioblastoma, natural killer T cell (NKT), lipid antigen, brain tumor, tumor immunity, glioma

## Abstract

Glioblastoma is an aggressive and deadly cancer, but to date, immunotherapies have failed to make significant strides in improving prognoses for glioblastoma patients. One of the current challenges to developing immunological interventions for glioblastoma is our incomplete understanding of the numerous immunoregulatory mechanisms at play in the glioblastoma tumor microenvironment. We propose that Natural Killer T (NKT) cells, which are unconventional T lymphocytes that recognize lipid antigens presented by CD1d molecules, may play a key immunoregulatory role in glioblastoma. For example, evidence suggests that the activation of type I NKT cells can facilitate anti-glioblastoma immune responses. On the other hand, type II NKT cells are known to play an immunosuppressive role in other cancers, as well as to cross-regulate type I NKT cell activity, although their specific role in glioblastoma remains largely unclear. This review provides a summary of our current understanding of NKT cells in the immunoregulation of glioblastoma as well as highlights the involvement of NKT cells in other cancers and central nervous system diseases.

## 1. Introduction

Glioblastoma (GBM) is the most common and aggressive primary central nervous system (CNS) tumor and is one of the deadliest human cancers [1]. Classified as grade IV glioma, GBM is a highly invasive tumor that is characterized by both molecular and morphological heterogeneity [2,3,4]. Standard treatment for GBM includes surgical resection, radiotherapy, and administration of the alkylating chemotherapeutic drug temozolomide (TMZ) [1,5]. Despite such intensive treatment, GBM has a median overall survival of 14.7 months, with only 6.8% of patients surviving beyond five years [6,7]. Although GBM comprises 1.4% of annual cancer incidences, it accounts for nearly double that in the proportion of cancer deaths [8]. Therefore, there is a pressing need for the development of additional therapeutic interventions. 

Immunotherapies have provided exciting clinical breakthroughs for a variety of other cancers, and they are an active area of GBM research. A variety of immunotherapies, including immune checkpoint blockade, vaccines, and adoptive T cell transfer, have been tested in the context of GBM [8,9,10]. Contrary to evidence from most preclinical glioma studies, immunotherapies have demonstrated little to no survival benefit to GBM patients thus far. Such resistance to immunotherapy is primarily attributed to the immunosuppressive GBM tumor microenvironment (TME). Our inability to effectively orchestrate immune-based interventions in GBM may be due to our incomplete understanding of the immunoregulatory mechanisms at play in GBM. For example, current frameworks of the GBM TME focus on macrophages, microglia, and regulatory T cells as the main immunosuppressive populations in GBM, but other cell populations may also contribute. Identifying additional immunoregulatory cell populations in the brain TME may lead to important insights and therapeutic targets. 

To date, most studies on the cytotoxic, regulatory, and/or immunosuppressive roles of T cells in GBM have focused on conventional T cells that recognize peptide antigens presented by major histocompatibility complex (MHC) molecules. However, natural killer T (NKT) cells, which recognize lipid antigens presented by non-classical class Ib MHC CD1d antigen-presenting molecules, have emerged as significant immune modulators in tumor immunity [11]. These cells may play an especially intriguing role in the regulation of brain cancer immune landscapes, as the brain is one of the most highly lipid-enriched organs in the body.

However, much is still unclear about how NKT cells contribute to the GBM TME, interact with existing immunotherapies, and may be manipulated to improve GBM outcomes. Therefore, the purpose of this review is to summarize our current understanding of the role of NKT cells in tumor immunity both generally and in the context of the immunological landscapes of GBM. As autoimmunity is considered to be the flip side of tumor immunity, we will also provide an overview of the role of NKT cells in multiple sclerosis, which is another central nervous system (CNS) disease mediated by T cells and regulated by NKT cells. Finally, we will consider the primary questions regarding the interactions between NKT cells and GBM that remain unanswered.

## 2. The Immune Landscapes of Glioblastoma

### 2.1. Neuroimmunity

Baseline immune functions in the brain parenchyma are carried out during homeostasis by microglia, which are the tissue-resident macrophages of the CNS. Microglia enter the CNS early in embryogenesis and maintain their CNS population by self-renewal, constituting 5–10% of total brain cells [12]. Microglia play important roles in neurodevelopment, synaptic remodeling, and phagocytosis of unwanted cells [13]. They also scan the entire brain parenchyma every few hours, sensing ATP to respond to CNS damage [14]. In response to inflammatory stimuli such as brain tumors, microglia rapidly change their morphology to an activated phenotype, which helps facilitate a response [9]. 

During homeostasis, cells originating in the peripheral immune system do not enter the brain parenchyma [15]. This physiological separation is primarily mediated by the endothelial cells of the blood–brain barrier (BBB), which are connected through a series of tight junctions that impede the direct entry of immunocytes and other molecules [16]. Furthermore, the vessel walls of the BBB are surrounded by a basement membrane and the glia limitans, which is an additional barrier formed by astrocytic end feet that wrap around the surface of the CNS vasculature to contribute to blood vessel impermeability [9,17]. 

This stringent partitioning initially misled researchers to believe that the CNS was “immune privileged”, which is a hypothesis that has since been disproven [13,18,19,20,21]. The segregation of the brain parenchyma and peripheral immunity is not unalterable; rather, there is flexibility in this separation driven by disease [15]. Adaptive and innate immune cells play important roles in the brain’s response to conditions including traumatic brain injury, Alzheimer’s, and brain tumors [18]. In pathological states including cancer, the integrity of the BBB is compromised, and this “leakiness” permits the entry of peripheral immune cells [22,23]. Under such inflammatory conditions, the expression of tight junction molecules in the endothelium decreases, and the glia limitans becomes discontinuous [24]. Additionally, the expression of adhesion molecules on the BBB increases, facilitating immune cell transmigration [25]. Once within the brain parenchyma, immunocytes follow chemokine gradients to localize to disease sites and exert their effects [17]. Thus, it is possible for peripheral immune cells to exert effects on intracranial tumors. 

The concept of CNS immune privilege has been further undermined by the discoveries of both the glymphatic system and the meningeal lymphatics [26,27,28,29]. Antigens are flushed out of the brain parenchyma in the cerebrospinal fluid (CSF) via the glymphatics [18]. This CSF can be sampled by immune cells, including T cells, in the functional meningeal lymphatic vessels [30]. Thus, the meningeal lymphatics can act as a site of antigen presentation and immune synapse [31]. Additionally, immune cells can readily extravasate through the meningeal vessels and infiltrate the parenchyma during pathological conditions [32]. Meningeal adaptive immunity has been shown to play a critical role in mediating T cell responses to GBM [33,34]. Moreover, the drainage of CSF from the meningeal lymphatics to the cervical lymph nodes is important for the trafficking of dendritic cells (DCs) and antigens from intracranial tumors to the periphery [34]. 

Evidence has emerged in recent years demonstrating the existence of resident memory CD8^+^ T cells (T_RM_) within brain parenchyma [35]. Early studies of T_RM_ cells within the brain observed the induction of these cells in response to viral CNS infections as well as long-term persistence of T_RM_ cells within the brain, which could provide protection against re-infection [36,37]. However, a recent study in mice demonstrated that non-CNS infections, such as influenza A, as well as peripheral immunizations, can also induce T_RM_ cell enrichment in the brain [38]. Intravenous injection of these mice with a fluorescent marker for CD45 before brain collection allowed for the distinction of tissue-resident vs. vascular cells, and the majority of post-infection or post-immunization antigen-specific T cells found in the brains of these mice were found to be located outside of the vasculature. Human postmortem brain samples from patients with and without CNS diseases were also shown to harbor CD69^+^ T_RM_ cells, which appeared to be localized to the perivascular spaces [39].

### 2.2. The GBM Microenvironment

The GBM TME includes a heterogenous mixture of cells, including not only tumor cells but also infiltrating peripheral immune cells, microglia, astrocytes, endothelial cells, fibroblasts, and pericytes [22,40]. GBM tumors are also highly invasive, integrating into the surrounding tissue where they can form electrochemical connections with nearby neurons to promote their own proliferation [41]. Additionally, GBM tumors are highly vascularized due to both angiogenesis and vascular co-option [22]. 

The GBM TME is characterized by immunosuppression. Tumor-associated macrophages (TAMs) and microglia make up the majority of infiltrating immune cells in GBM, comprising up to half of the tumor mass [42,43]. Of these, peripherally-derived monocytes and macrophages account for about 85%, with microglia comprising the remainder [42]. The phenotype of these cells can range from pro-tumorigenic to tumor-suppressive, but in GBM, TAMs are commonly skewed toward the immunosuppressive end of the continuum [44]. These immunosuppressive TAMs have high levels of STAT3 expression, which leads to the production of arginase that can interfere with CD3-mediated transduction of T cell signaling by depleting arginine from the microenvironment [45]. Expression of STAT3 in TAMs also leads to a downregulation of surface molecules that are involved in antigen presentation and co-stimulation, including major histocompatibility complex (MHC) Class II, CD40, CD80, and CD86, rendering them less effective at inducing T cell responses [46,47]. Furthermore, TAMs can support GBM growth by promoting angiogenesis, regulating neuroglial stem cell pools, and stimulating tumor growth and invasion [12,48].

Myeloid-derived suppressor cells (MDSCs) are also present in the GBM TME. Poly-morphonuclear (PMN)-MDSCs were shown to represent about half of CD45^+^ CD11b^high^ cells infiltrating human GBM tumors, although CD14^high^CD15^pos^ monocytic MSDCs are also significantly enriched in human GBM samples [49]. PMN-MDSCs can suppress the activity of antigen-specific T cells by producing reactive oxygen species [50]. Monocytic-MDSCs can further suppress T cell activity through the expression of nitric oxide synthase and arginase 1, which inhibits T cell proliferation, inhibits T cell receptor (TCR) signaling, and promotes T cell apoptosis [40]. MDSCs can also directly stimulate GBM angiogenesis and invasion [40]. The migration of MDSCs to the TME is chemokine dependent, and it was recently shown that blocking CXCL1 and CXCL2 can reduce the infiltration of MDCSs and improve prognosis in glioma models [51]. 

Neutrophils also play a generally immunosuppressive role in GBM tumors. Most research on neutrophils in GBM has focused on their interactions with anti-angiogenic therapies and their ability to act as prognostic indicators [22]. For example, a decreased ratio of neutrophils to lymphocytes in the peripheral blood of GBM patients during treatment with radiation and concomitant TMZ is a predictor of a prolonged overall survival [52]. On the other hand, high neutrophil levels in peripheral blood are associated with a positive response to bevacizumab, although high neutrophil levels within the TME are associated with bevacizumab resistance, suggesting a reprogramming of these cells upon invasion of the tumor tissue [53,54]. Recently, neutrophils were also implicated in the ferroptosis of GBM tumors, which leads to necrosis of tumor tissue and is associated with poorer survival, although the cause by which necrosis impacts survival is unknown [55]. Neutrophils were shown to transfer granules containing myeloperoxidase into GBM cells, inducing cytotoxicity through the iron-dependent buildup of lipid peroxides [55].

T cells can traffic into GBM tumors, but low proportions of T cells are found in most patients [56]. The majority of T lymphocytes in GBM tumors are CD8^+^ cytotoxic T cells, CD4^+^ helper T cells, and CD4^+^CD25^high^FoxP3^+^ regulatory T (Treg) cells [57,58]. GBM patients have high proportions of Tregs among CD4^+^ cells that contribute to immunosuppression in the TME, and the blockade of Treg function with anti-CD25 has been shown to prolong survival in glioma models [59,60]. On the other hand, extensive CD8^+^ T cell infiltration is associated with improved clinical prognosis in GBM [61]. However, T cells are often dysfunctional in GBM due to senescence, tolerance, anergy, or exhaustion [56]. While attempts have been made to combat T cell inhibition or exhaustion in GBM with anti-PD-1 and anti-CTLA-4 immune checkpoint blockade therapies, respectively, clinical trials have not demonstrated therapeutic efficacy [62]. T lymphocytes in GBM have also been shown to express additional inhibitory receptors including TIM-3, LAG-3, BTLA, 2B4, CD160, TIGIT, and CD39 [63]. Although the role of CNS T_RM_ populations in the context of brain tumors has been largely uncharacterized, it was recently demonstrated that mice re-challenged with GL261 syngeneic GBM tumors showed increased survival due to previous vaccination with irradiated GL261 cells infected with zika virus [64]. These mice harbored increased numbers of T_RM_ CD4^+^ cells compared to non-vaccinated mice, suggesting that the viral adjuvant to the cellular vaccination promoted tissue residence of CD4^+^ cells that could combat tumor recurrence.

GBM cells are active participants in creating and promoting their immunosuppressive microenvironment. GBM cells secrete immunosuppressive cytokines such as IL-10, transforming growth factor-β (TGF-β), and interleukin (IL)-6, which serve to suppress natural killer (NK) cell activity, suppress T cell activation and proliferation, induce T cell apoptosis, and skew TAMs to an immunosuppressive phenotype [25]. Importantly, GBM cells can also downregulate the expression of MHC on their surface due to the expression of IL-10 and TGF-β, hindering antigen presentation to CD8^+^ T cells [25]. Glioma cells further thwart the activity of T lymphocytes through their lack of CD80 and CD86 costimulatory molecule expression and their overexpression of PD-L1 co-inhibitory molecules [40]. On the other hand, the expression of STAT3 and indoleamine 2,3-dioxygenase (IDO) on GBM cells serves to increase the activity of Tregs and skew Th1 responses toward suppressive Th17 responses [56,65]. Furthermore, GBM tumor cells can suppress chemoattractants that promote lymphocyte trafficking to the site of the tumor [17]. Notably, GBM cells have altered lipid metabolism that leads to an aberrant expression of various lipids products [66]. These lipids have the ability both to act as antigens as well as manipulate immune cells in the TME, as discussed further below. 

## 3. NKT Cells

NKT cells are CD1d-restricted T cells that recognize lipid antigens rather than peptides. CD1d is a monomorphic class Ib MHC that is highly conserved among species. NKT cells are grouped into two types based on the TCR they express. Type I, or invariant, NKT (iNKT) cells express a semi-invariant TCRα chain comprised of Vα14Jα18 gene segments in mice and Vα24Jα18 segments in humans. The invariant TCRα chain pairs with a limited repertoire of TCRβ chains: Vβ2, 7, and 8 in mice and Vβ11 humans. Type II, or variant, NKT cells do not express the invariant TCRα but instead express variant TCRs similar to conventional T cells [67]. The nomenclature “type I” and “type II” used for the NKT cell groups is not analogous to Th1 and Th2 of CD4^+^ T cells. NKT cells in mice are either CD4^+^CD8^-^ or CD4^-^CD8^-^, while in humans, they can also be CD4^-^CD8^+^. NK cell receptors such as NK1.1 in mice and CD161 and CD56 in humans can be expressed on a subset of NKT cells, but they cannot be used to identify NKT cells [68].

NKT cells have several unique characteristics that make them distinct from conventional T cells. Firstly, the two subsets of NKT cells recognize different sets of antigens with mostly non-overlapping repertoires. For example, a representative antigen for type I NKT cells, α-galactosylceramide (α-GalCer, KRN7000), is not recognized by type II NKT cells, while a representative antigen for type II NKT cells, sulfatide (3-*o*-sulfogalactosylceramide), is not recognized by type I NKT cells in mice [69]. Secondly, a single TCR of NKT cells can recognize antigens with a wide variety of structures. This is significantly different from TCRs of conventional T cells, which are restricted by class I or class II MHC. Although single TCRs of conventional T cells are known to be able to recognize peptides with some amino acid substitutions, called altered peptide ligands, that induce different qualities of activation in conventional T cells [70], the variety of lipid antigen structures recognized by a single TCR of NKT cells is much larger. Similarly to altered peptide ligands, lipid antigens with different structures induce significantly different cytokine profiles in activated NKT cells [71,72]. For instance, in activated type I NKT cells, α-GalCer induces both Th1 and Th2 cytokines, while C20:2 and OCH induce more Th2 skewed cytokines [73], and α-C-GalCer and 7DW5-8 induce highly Th1 skewed cytokines [74,75]. 

The semi-invariant TCR-α chain interacts solely with an antigen-CD1d complex, and the TCR-β chain does not react with antigens, but with a CD1d molecule [76]. Thus, although the frequency of NKT cells in most organs is significantly lower than that of conventional T cells (e.g., 1% vs. 40–45% in mouse spleens), the precursor frequencies of NKT cells are much higher than those of conventional T cells [77]. In addition, a significant fraction of NKT cells display an activated/memory phenotype (CD44^hi^CD69^+^) in naïve mice and have pre-formed mRNA of cytokine genes, which allows them to produce large amounts of various cytokines within hours of activation [78]. Many NKT cells are also tissue resident [79,80] as they constitutively express CD69, a tissue homing receptor. In fact, it is reported that 10% of resident T cells (after removing peripheral blood) are type I NKT cells in whole mouse brain, although their absolute location in the tissue is not known [81].

Similar to CD4^+^ T helper cells, type I NKT cells have been shown to have functional subsets, at least in mice. These subsets can be identified by transcription factor expression through the combination staining of PLZF, T-bet, and ROR-γt [82]. NKT1, NKT2, and NKT17 are considered to be counterparts of Th1, Th2, and Th17 CD4^+^ T cell subsets, respectively, although it should be noted that all type I NKT cell functional subsets produce Th2 cytokine IL-4. In addition to these three functional subsets, NKTreg [83], NKT10 [84,85], and NKT_FH_ [86,87] subsets have also been reported. There may be more subsets based on single-cell transcriptomic analysis [88]. Recent single cell RNAseq analysis of human peripheral blood NKT cells suggests that there may not be such functional subsets in humans [89]. However, it has been shown that CD4^+^ and CD4^-^CD8^-^ type I NKT cells have different cytokine profiles, with CD4^+^ type I NKT cells predominantly producing IL-4 and CD4^-^CD8^-^ type I NKT cells producing both IL-4 and IFN-γ [90,91]. In contrast to type I NKT cells, almost nothing is known about the functional subsets of type II NKT cells. A summary of the characteristics of NKT cells can be found in Table 1.

## 4. NKT Cells in Tumor Immunity

### 4.1. Type I NKT Cells

Type I NKT cells have been shown to promote tumor immunity in multiple preclinical models. Type I NKT cell-deficient Jα18 knockout (KO) mice and NKT cell-deficient CD1d KO mice were both shown to have faster tumor growth compared to wild-type mice with intact NKT cells in a fibrosarcoma model, suggesting that type I NKT cells facilitate tumor immunosurveillance [92]. This enhancement of immunosurveillance was demonstrated to be dependent upon IFN-γ secretions from both NKT cells and CD8^+^ T cells in adoptive transfer experiments [92]. The immunoenhancing role of type I NKT cells in tumor immunity is also evident in multiple spontaneous tumor models. Type I NKT-deficient Jα18 KO mice are more susceptible to spontaneous fibrosarcoma development after methylcholanthrene injection [93], and they also show accelerated tumor onset in a TRAMP prostate tumor model [94].

When type I NKT cells are stimulated by exogenous agonistic antigens such as α-GalCer, activated type I NKT cells exert their anti-tumor activity through two mechanisms: direct killing of CD1d-expressing tumor cells and facilitation of anti-tumor immune responses through induction of effectors cells, including NK cells and CD8^+^ T cells, via the production of cytokines such as IFN-γ (Figure 1).

When α-GalCer is presented by DCs in the context of CD1d, an interaction between CD40 and CD40L induces the maturation and activation of DCs such as the production of IL-12 [95,96]. CD1d engaged with a TCR can also transduce a signal to induce the expression of IL-12 in DCs [97]. IL-12 induces the activation of NK cells, CD8^+^ T cells, and Th1 cell differentiation. The interaction between type I NKT cells and CD8^+^ DCs, in the context of presentation of agonistic glycolipid antigens by CD1d, also facilitates the cross-presentation of protein antigens by CD8α^+^ DCs. Thus, α-GalCer and its analogs have been shown to provide a strong adjuvant effect to vaccine-driven induction of CD8^+^ T cells [96]. For example, intravenous co-injection of a protein antigen with α-GalCer induces stronger antigen-specific CD8^+^ T cell responses than protein antigen alone, which does not induce any CD8^+^ T cell responses [98]. It also has been shown that α-GalCer can act as a strong adjuvant for irradiated tumor cell vaccines when tumor cells are pulsed with α-GalCer prior to irradiation, regardless of the expression of CD1d on tumor cells [99]. In the GL261 syngeneic mouse GBM model, irradiated GL261 pulsed with α-GalCer and doxicyclin induced CD8^+^ T cell-mediated anti-tumor response in both prophylactic and therapeutic settings (6 days after tumor implantation) [100]. The efficacy of such vaccination was further enhanced when combined with Treg depletion.

In most preclinical studies with α-GalCer and its analogs in mouse tumor models, α-GalCer has been administered when tumor cells were inoculated, with a few exceptions. This may be because delaying α-GalCer administration after tumor inoculation will not provide protective effects, suggesting that type I NKT cell agonists as single agents may have limited effects on tumor growth in a therapeutic setting.

Type I NKT cells also interact with other myeloid cell populations and can manipulate their functions. α-GalCer activated type I NKT cells increase immunostimulatory macrophage populations in tumor-bearing mice [101,102] and reduce the suppressive activity of MDSCs [103,104,105,106,107]. When interacting with type I NKT cells by presenting α-GalCer, MDSCs are promoted to produce pro-inflammatory cytokines [104] and differentiate into antigen-presenting cells [103,106]. Human type I NKT cells have been shown to have less susceptibility to suppression by CD15^+^ MDSCs than conventional T cells in head and neck cancer patients [108]. Neutrophils are another myeloid cell population that potentially interacts with type I NKT cells. Serum amyloid A-1 (SSA-1) produced in melanoma patients as a result of inflammation induced an increase in IL-10 producing immunosuppressive neutrophils. SSA-1 simultaneously induces the expression of CD1d in neutrophils, which in turn facilitates interaction with and activation of type I NKT cells independently of an exogenous antigen [109]. The interaction between type I NKT cells and immunosuppressive neutrophils diminishes IL-10 production and enhances IL-12 production by neutrophils. Reversely, high concentrations of neutrophils were shown to suppress the activation and cytokine production of type I NKT cells in both mice and humans [110]. Overall, the outcomes of type I NKT–myeloid cell interactions appear to be context dependent.

Evasion of immune surveillance is considered an important hallmark of cancer, and a reduced number and function of effector immune cells can contribute to such defective immunological monitoring of tumors [111]. In cancer patients, type I NKT cells have been reported to have defects in their number and functions in the peripheral blood [112,113,114,115,116]. A low number of type I NKT cells is reported to be a predictor of disease prognosis for head and neck cancer patients [117]. Frequently, IFN-γ production ability is diminished in peripheral type I NKT cells in cancer patients, although IL-4 production and lytic activity is not changed [118]. The reduced IFN-γ productivity of type I NKT cells in cancer patients is in part attributed to defective interactions between type I NKT cells and DCs [119]. In colorectal cancer patients, a high number of infiltrated type I NKT cells in tumor tissue is reported to be a predictor of a better prognosis [120]. These clinical data suggest that type I NKT cells play a role in tumor surveillance in human cancer patients. 

α-GalCer has been tested as a cancer treatment in numerous clinical trials [117,121,122,123]. In most studies, α-GalCer was used as a single agent, either as a solo drug or pulsed on autologous DCs. Although treatments with α-GalCer expanded circulating type I NKT cell numbers and/or increased IFN-γ levels, no partial or complete remission was observed in most trials. Studies treating head and neck cancer patients with α-GalCer-pulsed APCs combined with adoptive transfer of activated type I NKT cells reported a partial response in three out of seven patients in one study and five out of 10 patients in another study [124]. Another study in 17 lung cancer patients treated with α-GalCer-pulsed APCs reported that patients with increased levels of IFN-γ producing cells had significantly longer median survival compared to patients without increased IFN-γ producing cells [124]. 

While Type I NKT cells directly lyse cancer cells expressing CD1d, most solid cancers are negative for CD1d expression. However, neuroblastoma retains CD1d expression and can be recognized by type I NKT cells [125]. In a xenograft model of human neuroblastoma, adoptive transfer of human type I NKT cells with 7DW8-5 administration reduced tumor burden and prolonged the survival of mice [125]. Interestingly, NKT cells also selectively killed CD1d^+^CD68^+^ TAMs. The combination treatment of adoptively transferred type I NKT cells and α-GalCer administration was also recently reported to effectively control CD1d^+^ human GBM in a xenograft model, which is discussed below.

As mentioned above, type I NKT cells enhance tumor immunity in most preclinical mouse tumor models. However, it should be noted that in several models, type I NKT cells were reported to contribute to the suppression of tumor immunity. In a B16SL4 model of liver metastasis, type I NKT cells suppressed NK cell-mediated immunosurveillance through an IL-10 dependent mechanism [126]. The suppression of tumor immunity by type I NKT cells in this model cannot be explained by the location of tumors within the liver, as therapeutic effects of α-GalCer treatment were shown in a B16 liver metastasis model [127] and liver CD4^-^CD8^-^ NKT cells were reported to be highly potent inducers of anti-tumor immune responses among NKT cell subsets in spleens, livers, and thymus [128]. Furthermore, Jα18 KO mice, which are deficient in type I NKT cells, developed fewer liver metastases compared to wild-type mice [126]. Another study in a Burikitt’s-like B cell lymphoma model also found that Jα18 KO mice had significantly lower tumor burden compared to wild-type and CD1d KO mice [129]. In this model, type I NKT cells seem to suppress CD8^+^ T cell-mediated immunosurveillance. Type I NKT cells also suppress tumor immunosurveillance in APC^Min/+^ mice, which is a model of spontaneous colon cancer [130]. Type I NKT cells infiltrated polyps in this model produced IL-10 and IL-17, although the number of type I NKT cells in the lamina propria and spleen did not change between APC^Min/+^ and wild-type mice, suggesting that the quality but not quantity of type I NKT cells was altered. In this model, activation of type I NKT cells with various agonistic antigens led to different outcomes depending on the character of antigen used and duration of NKT cell activation [102]. Treatment with α-GalCer, which induces both Th1 and Th2 cytokines, reduced the polyp burden associated with enhanced CD8^+^ T cell responses. In contrast, long-term treatment with C20:2, which induces more Th2 skewed cytokine production, enhanced polyps, while short-term treatment with C20:2 showed an opposing effect to suppress polyps. Interestingly, long-term treatment with α-C-GalCer, which induced a highly Th1 skewed cytokine profile, increased polyp numbers. While the underlining mechanisms determining the role of type I NKT cells in tumor immunity remain unclear, studies with the APC^Min/+^ model suggest that these mechanisms are dependent on the antigens type I NKT cells recognize and the duration of NKT activation.

### 4.2. Type II NKT Cells

In contrast to type I NKT cells, type II NKT cells have been reported to have an immunosuppressive function in multiple mouse transplantable syngeneic models. In those models, type II NKT cells suppress CD8^+^ T cell-mediated tumor immunosurveillance through the production of IL-13. IL-13, together with TNF-α, induces TGF-β production in myeloid cells, and TGF-β suppresses CD8^+^ T cells [131,132,133,134]. The activation of type II NKT cells with sulfatide was also reported to suppress tumor immunity in a subcutaneous 15-12RM fibrosarcoma model and a CT26 colon carcinoma lung metastasis model, as sulfatide treatment of mice facilitated tumor growth [135]. Similar results were also reported in a B cell lymphoma model [136]. In line with observations in mouse models, in myeloma patients, type II NKT cells produce IL-13 when they are stimulated with cancer-derived lyso-phosphatidyl choline [137]. Thus, it is likely that type II NKT cells also have an immunoregulatory function in humans.

### 4.3. Interactions between Two Types of NKT Cells and Tumor Immunity 

The two types of NKT cells not only have generally opposing functions in the regulation of tumor immunity, but they also cross-regulate each other. This relationship between the two types of NKT cells was revealed in a study in which each type was specifically stimulated with an agonistic antigen both in vivo and in vitro [135]. In vitro, the stimulation of type II NKT cells with sulfatide suppressed the proliferative response of type I NKT cells induced by α-GalCer. Type I NKT cells stimulated in vivo by the administration of α-GalCer significantly reduced tumor burden in the subcutaneous 15-12RM fibrosarcoma model and the CT26 colon carcinoma lung metastasis model, while type II NKT cell stimulated in vivo by the administration of sulfatide significantly increased tumor burden. When both types of NKT cells were simultaneously stimulated in vivo by the co-administration of α-GalCer and sulfatide, the protective effects of α-GalCer were significantly reduced. Simultaneously, the tumor-enhancing effects of sulfatide were also reduced. Therefore, this suggests that the two types of NKT cells cross-regulate each other [138]. The mechanisms of counteraction of type II NKT cells against type I NKT cells were proposed by Halder et al. [139]. In a ConA-induced hepatitis model, in which type I NKT cells cause inflammation in the liver, sulfatide-activated type II NKT cells were shown to induce the anergy of type I NKT cells by activating plasmacytoid DCs to produce MIP-2 and IL-12 that recruit type I NKT cells into the liver to induce anergy. Whether the mechanism proposed in the ConA-induced hepatitis model is active in tumor models is currently unclear. Further studies are required to elucidate the mechanism of cross-regulation between the two types of NKT cells in tumor settings.

The interaction between the two types of NKT cells also can affect the role of Foxp3^+^CD4^+^ Treg cells. In a subcutaneous CT26 tumor model, suppressing Treg functions with anti-CD25 treatment induced the rejection of tumors in both wild-type mice and type I and II NKT cell-deficient CD1d KO mice [140,141]. In contrast, this treatment had no effect on tumor growth in Jα18 KO mice, which have only type II NKT cells. However, combining anti-CD25 treatment with CD1d blockade, thereby inhibiting the activation of type II NKT cells, led to tumor rejection in Jα18 KO mice, indicating that both Treg and type II NKT cells are suppressing tumor immunity in Jα18 KO mice. Additionally, Jα18 KO mice became responsive to anti-CD25 treatment upon the adoptive transfer of type I NKT cells. Thus, the counteraction of type II NKT cell suppressive functions with type I NKT cells leaves Tregs a dominant immunosuppressive cell. Therefore, it is plausible that the reductions in type I NKT cell numbers and IFN-γ productivity frequently seen in cancer patients [112,113,114,115,116,117,118] help to explain why Treg-targeted therapies have not been successful in patients, as type II NKT cells may remain largely unchecked and able to exert immunosuppressive effects.

## 5. Lipids in Brain Immunology and GBM

The brain is the second-most lipid enriched tissue in the body following the adipose tissue, and approximately half of the dry weight of brain tissue is lipids [142]. The major lipids classes in the brain are sphingolipids, cholesterol, and glycerophospoholipids. Lipids are not only major components of the cell membrane; they are also involved in multiple functions of the central nervous system (CNS), including development and signaling. Since tumor cells have abnormal metabolism, lipid composition in glioma is significantly altered in comparison to normal human brain [66,143]. Some lipids, such as gangliosides, are shed from tumor cells [144] and can induce immune responses such as antibody production (Figure 2) [145].

The antibody production is presumably elicited by NKT_FH_ providing help to B cells [86,87]. Thus, lipids produced by glioma are immunogenic. On the other hand, glycosphingolipids produced by glioma cells play critical roles in maintaining their tumorigenicity. For instance, the gangliosides GD2 and GD3 increase the motility, invasiveness, and proliferation of a GBM cell line, U251, and primary mouse astrocytes [146,147]. GD3 is overexpressed in GBM stem cells and GBM tissues compared to normal brain tissues and plays a critical role in maintaining stem cell properties [148]. It was reported that the expression of glucosylceramide synthase was increased in a TMZ-resistant human GBM cell line T98G and that the inhibition of glucosylceramide synthase increased the sensitivity to the drug [149]. Glycosphingolipids synthesized by gliomas can also regulate immune cells. GD3 was reported to induce the apoptosis of activated T cells [150], though interestingly, GD3-mediated induction of apoptosis was not observed against resting T cells. Overall, the lipid metabolites that result from abnormal lipid metabolism in glioma contribute to the maintenance of tumorigenic properties of glioma cells, modulate immune cells in the tumor microenvironment, and serve as potential targets that can be recognized by the immune system (Figure 2).

## 6. NKT Cells in GBM

Among the major types of lipids, sphingolipids and glycerophospholipids can be presented by CD1d and recognized by NKT cells [151]. However, there are only a handful of studies that have attempted to investigate the connection between NKT cells and gliomas. Dhodapkar and Steinman were the first to report that some GBM cells from patients express CD1d and can be killed when they present α-GalCer to patients’ NKT cells in vitro, while they were not lysed without adding exogenous antigens. They also reported that GBM patients preserved the number and IFN-γ producing ability of NKT cells [152], contrasting observations in patients with other types of solid cancers who have reduced numbers of NKT cells with diminished IFN-γ production [118]. However, this study did not address whether NKT cells infiltrate into GBM tissues. This question has been examined by three independent studies that have provided conflicting data. A study of seven GBM patient tumor samples showed that nearly half of the GBM tumors examined were infiltrated by CD4^+^ T cells expressing CD56, an NK cell marker, and the frequency of CD56^+^CD4^+^ T cells was much higher than in meningiomas and metastatic lung cancers [153]. However, these cells did not react with α-GalCer-loaded CD1d tetramers and also did not respond to fixed CD1d-transfected B cell lymphoblastoma, indicating that they were not type I NKT cells or CD1d-reactive NKT cells, respectively. Similarly, a recent study reported that type I NKT cells could not be detected by flow cytometry in TILs of 15 GBM patients [154]. The authors reported that GBM cells from 10 out of 15 patients expressed CD1d. In contrast, a third study reported that IL-6^+^IL-10^+^ immunosuppressive type I NKT cells were detected in GBM tissues [155]. These IL-6^+^IL-10^+^ type I NKT cells showed the reduced expression of IFN-γ, perforin, and Fas-L. The induction of IL-6^+^IL-10^+^ type I NKT cells was mediated by miR-92a expressed by GBM cells. It is not clear whether these type I NKT cells in tumor tissue migrated from peripheral blood or were residents of the brain, as NKT cells are considered to be tissue-resident T cells, and it was reported that 10% of brain-resident T cells are type I NKT cells in C57BL/6 mice, although the exact location in the brain was not identified [81]. It should be taken into consideration that the frequency of type I NKT cells in peripheral blood of humans varies over a two-log difference (0.01–3%) among individuals, and the frequency of type I NKT cells in tissues may show similar amounts of variation. Hence, whether NKT cells can be found in tumor tissues may be a reflection of an individual’s frequency of NKT cells in circulation. However, it may not be critical for NKT cells to infiltrate into tumor tissues, as they can exert anti-tumor functions by producing cytokines to modulate effector T cells that are primed in draining lymph nodes. It remains unknown whether NKT cell levels correlate to response rates to immunotherapies. Thus, the role of NKT cells in natural immunosurveillance and immunotherapy of glioma needs to be further elucidated.

The potential of NKT cells to facilitate tumor immunity in preclinical GBM models has been demonstrated in α-GalCer stimulation studies (Figure 3).

As mentioned above, in the GL261 syngeneic orthotopic GBM model, the combination of irradiated tumor cells, α-GalCer, and doxicyclin was shown to effectively induce CD8^+^ T cell mediated anti-tumor immune responses [100], and this efficacy was further enhanced by Treg depletion through anti-CD25 treatment. In a C6 rat orthotopic GBM model, a DC vaccine pulsed with both tumor-derived exosomes and α-GalCer induced the prolonged survival of tumor-bearing rats [156]. The effect of this vaccine was accompanied by an increase in tumor-specific CD8^+^ T cell responses and increased cytokine levels in peripheral blood.

Type I NKT cells can also be used as effector cells for adoptive immunotherapy if tumor cells express CD1d, as NKT cells recognize tumor cells and lyse them when tumors present α-GalCer [154]. In a U251 orthotopic xenograft model of GBM, mice that received intracranially transferred human type I NKT cells and α-GalCer showed significantly reduced tumor burden and prolonged survival compared to mice that received either NKT cells or α-GalCer alone. Along with the previously mentioned studies, this suggests that if NKT cells are optimally stimulated with an agonistic lipid antigen, they can facilitate anti-GBM immunity. Altogether, these preclinical studies in various GBM models, as well as neuroblastoma models, suggest that NKT-targeted immunotherapy has the potential to improve the outcome of nervous system cancers (Figure 3).

## 7. NKT Cells in Multiple Sclerosis

In addition to brain cancer, a well-studied pathology in the CNS is multiple sclerosis (MS), which is an autoimmune disorder of the CNS causing acute and chronic inflammation. Although the cause of this disease is not clear, auto-reactive CD4^+^ T cells are believed to be involved in the pathology. In this autoimmune disease, NKT cells have been implicated in the regulation of immune responses (reviewed in detail in [157,158,159]). The quantitative changes in peripheral blood levels of NKT cells in MS patients have been conflicting, as some studies reported a reduction [113,160,161,162], one study reported an increase [163], and another study reported no change [164]. However, there is consensus among the reports that there are qualitative changes in NKT cells in MS. De Biasi et al. reported that type I NKT cells in the peripheral blood of MS patients with secondary progress have increased pro-inflammatory cytokine (IL-17 or TNF-α) production [165]. A study of type I CD4^+^ NKT cell clones established from patients with remission reported that IL-4 production was higher than in cells from healthy donors [160]. These studies suggest that the cytokine profile of type I NKT cells changes depending on disease status.

A mouse model of MS is experimental autoimmune encephalomyelitis (EAE), in which disease is induced by the immunization of mice with myelin antigen and an adjuvant such as complete Freund’s adjuvant. In this model, both type I and type II NKT cells can be detected in the CNS [166,167]. The role of NKT cells as assessed by using NKT cell-deficient mice with EAE has been conflicting. In NKT cell-deficient CD1d KO mice, the disease showed no change [168,169,170] or was exacerbated [171]. Similarly, studies showed no change [169] or disease exacerbation [172] in type I NKT cell-deficient Jα18 KO mice compared to NKT cell intact mice. However, type I NKT cells were shown to be protective when Vα14Jα18 transgenic (Tg) mice were compared to wild-type mice or when type I NKT cells from Vα14Jα18 Tg mice were adoptively transferred to wild-type mice [173]. The protection was mediated by the suppression of IFN-γ production by auto-reactive T cells.

Moreover, the activation of NKT cells with agonistic lipid antigens also provided diverse outcomes in EAE models. In most studies, treatment of mice with α-GalCer or its analog ameliorated the disease, and this was associated with an enhanced production of Th2 cytokines or reduction of pathologic Th1/Th17 cytokines. Multiple mechanisms have been proposed for the protective activity of stimulated type I NKT cells. The protective role of Th2 cytokines produced by type I NKT cells was suggested by a study using an α-GalCer analog OCH, which induces a more Th2 skewed cytokine profile in activated type I NKT cells [174], and by a study combining CD86 blockage with α-GalCer, which promotes Th2 cytokine production by type I NKT cells [175]. It has also been suggested that type I NKT cells modulate T cell responses through myeloid cell populations such as MDSCs [176], DCs [177], and macrophages [172].

Not only type I NKT cells but also type II NKT cells can regulate T cell responses against CNS antigens. Sulfatide is one of the major cell membrane components of myelin sheath, and it specifically activates type II NKT cells in mice. In vivo administration of sulfatide increased type II NKT cells in the brain and protected mice from EAE [166]. The protection was mediated through the induction of tolerogenic DCs, which could transfer the protection by adoptive transfer of the tolerogenic DCs into naïve recipients. Sulfatide treatment also suppressed the expression of CD1d and B7 molecules in microglia and reduced the number of type I NKT cells in the brain [167].

Altogether, insights from MS and EAE studies suggest that NKT cells have the ability to regulate T cell-mediated immune responses against CNS antigens. However, the directions of immune responses regulated by NKT cells are diverse, and how these directions are determined remains unclear.

## 8. Further Questions Regarding NKT Cells in GBM Immunity

The studies of MS provide outstanding questions that need to be addressed to understand the role of NKT cells in brain cancer. First, it will be important to determine the identity and nature of antigens recognized by NKT cells in CNS and/or glioma. As mentioned above, sulfatide is one of the glycolipids in CNS known to be recognized by NKT cells, but sulfatide is a group of glycolipids that contain multiple species. For instance, the rat cerebellum contains more than 15 species of sulfatide that have different structures of acyl chain and/or sphingosine chain [178]. However, the sulfatide species that have been studied for antigenic activity against NKT cells are very limited and have mostly been tested in a d18:1/24:1 ceramide [179,180]. The composition of each lipid species changes in tumor cells compared to normal cells [66,143], because the availability of lipid components is affected by metabolic status. Numerous studies of α-GalCer analogs have demonstrated that changes in acyl chain and/or sphingosine chain structure significantly alter the quality and quantity of activated NKT cells [181]. Thus, it is highly likely that different species of glycosphingolipid have different activity against NKT cells. However, such a structure–function study of antigens for NKT cells conducted for species other than α-GalCer are very limited, and the information obtained from α-GalCer may not be applicable to other glycosphingolipids [182,183]. Thus, detailed lipid profiling and the examination of antigenic activity of each lipid species are required to understand the function of NKT cells in glioma.

It is also important to note that some types of lipids that can bind to CD1d serve as antagonists that occupy CD1d but are not recognized by the TCRs of NKT cells, thus preventing agonistic lipids from being loaded onto CD1d. One such example is sphingomyelin, which is one of the most abundant sphingolipids [184,185]. GD3 is also reported to bind with CD1d, but it is not recognized by NKT cells [186]. This ganglioside outcompetes α-GalCer for binding to CD1d and suppresses type I NKT cells. On the other hand, another study demonstrated that GD3 pulsed-DCs activated type I NKT cells in vivo [187]. This discrepancy may be a result of using GD3 with different ceramide structures, as those two studies used GD3 derived from cells from different tissue origins, with the former using ovarian cancer and later using melanoma as a source of the ganglioside.

Revealing the functions of NKT cells in the normal brain will also be helpful in understanding the role of NKT cells in glioma, as type I NKT cells comprise approximately 10% of T cells in the brain, although their absolute location is not understood [81]. Emerging evidence has clearly demonstrated meningeal lymphatic drainage between the brain parenchyma and the cervical lymph nodes. Thus, there is close communication between the brain parenchyma, surrounding non-parenchymal tissues, and brain resident NKT cells, and this may play a role in the immune responses that occur in the brain parenchyma during pathological conditions.

Finally, it will be important to elucidate which cells are presenting antigens to NKT cells in the CNS. Microglia express CD1d, and it has been shown that sulfatide-driven activation of type II NKT cells can induce the tolerogenic activity of microglia by downregulating co-stimulating molecules [167]. Since sulfatide is highly abundant in CNS, it remains puzzling as to why endogenous sulfatide does not activate the pathways that suppress autoreactive T cells in EAE. It remains unclear whether this pathway is active in glioma.

Seeking answers to these questions may lead to important new understandings of how NKT cells are involved in the immune responses against CNS tumors. It may also provide new, specific targets for therapeutic intervention. For example, it may be possible to use drugs, exogenous lipids, or ex vivo engineered immune cells to target or induce specific lipid populations in GBM in order to manipulate NKT cells and downstream effector cells. It will also be important to determine whether type II NKT cells play an immunosuppressive role in GBM and/or cross-regulate the activity of type I NKT cells in the context of the brain. If so, inhibiting type II NKT cells may be key for reducing immunosuppression in the GBM TME and facilitating the effectiveness of other types of immunotherapies, such as CAR-T cell therapies and vaccines. Tipping the balance between type I and type II NKT cells to a more type I NKT cell dominant profile, perhaps through the administration of exogenous antigens for type I NKT cells, may provide therapeutic benefit to GBM patients. Furthermore, promoting the anti-tumorigenic activity of type I NKT cells via adoptive transfer or antigen administration may prove useful in combination with other GBM-targeted immunotherapies. Importantly, single agent treatment with NKT cell-targeting drugs will likely be insufficient to create an immune response against GBM; rather, it will be important to consider combining NKT-targeted therapy with other treatments such as vaccines, chemotherapy, and radiotherapy. For example, stimulated type I NKT cells may have the ability to act as powerful adjuvants to GBM-targeted vaccines, given their ability to stimulate cross-presentation of antigen by DCs. Overall, clarifying the role of NKT cells in GBM and harnessing those insights to potentiate effective therapies may lead to meaningful breakthroughs in GBM treatment.

## 9. Conclusions

In this review, we have highlighted the role of NKT cells in the immunoregulation of GBM, as well as summarized the opposing mechanisms by which type I and type II NKT cells are involved in the regulation of other types of cancers and CNS disease. While it is clear that Type I NKT cells can facilitate the killing of GBM cells in vitro and that α-GalCer can act as an adjuvant to increase the survival of mice with intracranial tumors, it is still uncertain whether type I NKT cells are present in GBM tumors and exert anti-tumor effects in situ. There is also a lack of research on the implications of type II NKTs in GBM, but the ability of these cells to indirectly suppress CD8^+^ T cell activity in the context of other tumors indicates that they may be important immunosuppressive cells in GBM as well. It is our hope that a stronger understanding of the immunoregulatory roles of type I and type II NKT cells in GBM will provide us with a more thorough framework upon which to develop effective immune-based treatments for this terrible disease.

## Figures and Tables

**Figure 1 cells-10-01641-f001:**
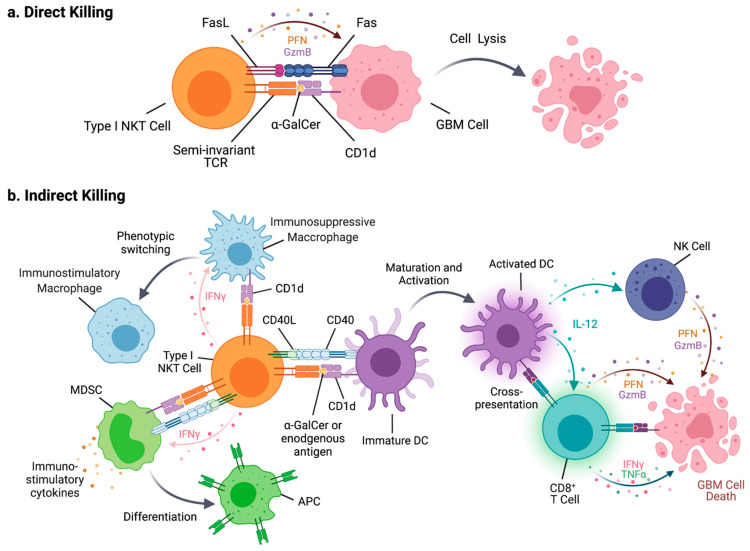
Potential mechanisms of type I NKT cell-mediated glioblastoma cell lysis. (**a**) Type I NKT cells can directly lyse GBM cells expressing CD1d presenting exogenous lipid antigens such as α-GalCer via Fas/FasL interaction and perforin (PFN) and granzyme B (GzmB) release. (**b**) Type I NKT cells can indirectly induce GBM cell lysis when presented with lipid antigens via CD1d on DCs. Through interaction with type I NKT cells, DCs mature and activate, cross-present antigens to CD8^+^ T cells, and secrete IL-12 to induce CD8^+^ T cells and NK cells to directly lyse GBM cells. Additionally, activated type I NKT cells can induce macrophages to switch from an immunosuppressive to an immunostimulatory phenotype when macrophages present lipid antigens in the context of CD1d. Type I NKT cells can interact with MDSCs via TCR-CD1d interaction and CD40–CD40 ligand (CD40L) interaction to induce MDSC secretion of pro-inflammatory cytokines and differentiation into antigen-presenting cells (APCs).

**Figure 2 cells-10-01641-f002:**
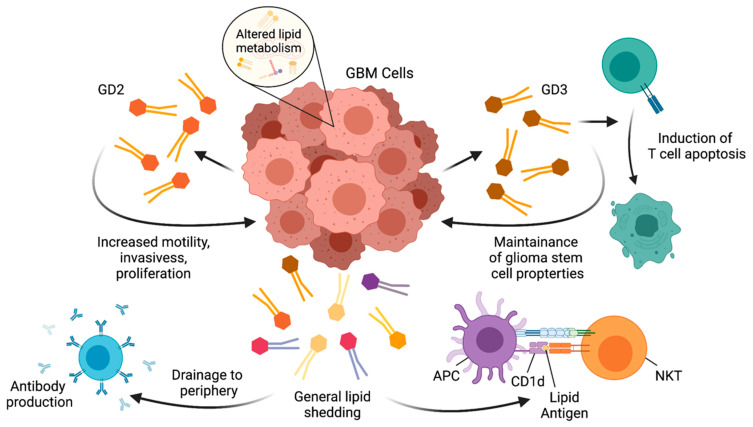
Altered lipid metabolism in GBM promotes cancer properties and interacts with immune cells. Alterations in lipid metabolism in GBM cells causes an increased production of GD2, which increases motility, invasiveness, and proliferation of GBM cells, as well as GD3, which is involved in the maintenance of glioma stem cell properties and can induce T cell apoptosis. Additional lipids shed from GBM cells can drain to the periphery, where they can act as antigens and stimulate antibody production. Lipid antigens derived from GBM cells can also be presented to NKT cells in the context of CD1d.

**Figure 3 cells-10-01641-f003:**
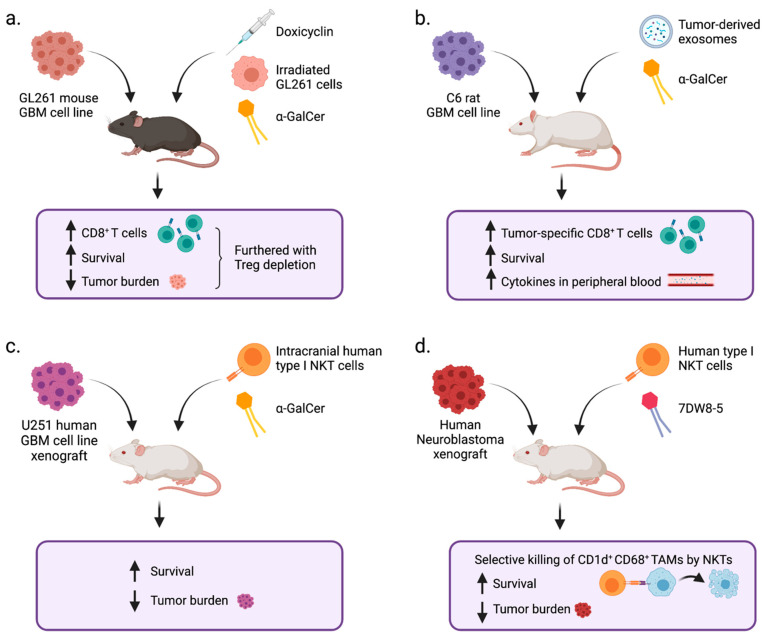
Summary of studies involving NKT cell agonists or NKT cells to treat nervous system cancers. (**a**) Mice implanted with the GL261 mouse GBM cell line were treated with doxycyclin, irradiated tumor cells, and α-GalCer. Mice showed an increase in CD8^+^ T cells and a decrease in tumor burden with treatment, and this was furthered by additional treatment with Treg depletion via anti-CD25. (**b**) Rats implanted with C6 rat GBM cell line were treated with tumor-derived exosomes and α-GalCer. Rats showed an increase in tumor-specific CD8^+^ T cells, increased survival, and increased cytokines circulating in peripheral blood with treatment. (**c**) Mice implanted with U251 human GBM cell line xenograft tumors were treated with intracranial adoptive transfer of human type I NKT cells and α-GalCer. Mice showed an increase in survival and a decrease in tumor burden with treatment. (**d**) Mice implanted with human neuroblastoma xenografts were treated with adoptive transfer of human type I NKT cells and 7DW8-5. Mice showed selective killing of CD1d^+^CD68^+^ TAMs by NKTs, increased survival, and decreased tumor burden with treatment.

**Table 1 cells-10-01641-t001:** Characteristics of type I and type II NKT cells.

	Type I NKT	Type II NKT
Antigen-presenting molecule in immunological synapse	CD1d	CD1d
TCRα	Vα14 Jα18 (mice)	Diverse
	Vα24 Jα18 (humans)	
TCRβ	Vβ2, 7, 8 (mice)	Diverse
	Vβ11 (humans)	
Functional subsets	NKT1, NKT2, NKT17	Not Known
	NKT10, NKT_FH_, NKTreg	
Representative antigen recognized	α-GalCer	Sulfatide

Role in cancer (with exceptions)	Enhance immunity	Suppress immunity
Major mechanism of action	Production of IFN-γ, induction of IL-12 production by DCs and CD40–CD40L interaction	Production of IL-13 to induce TGF-β production by myeloid cells

## Data Availability

Not applicable.

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
