# Peer review of "The Role of NKT Cells in Glioblastoma"

_cells, 2021, doi:10.3390/cells10071641_

Round 1

Reviewer 1 Report

Ref Manuscript: cells-1257291
Title: The Role of NKT Cells in Glioblastoma
Journal: Cells

The article entitled: “The Role of NKT Cells in Glioblastoma” by Brettschneider & Terabe, is a review article providing a summary of the current understanding of NKT cells in the immunoregulation of glioblastoma and the involvement of NKT cells in other cancers and central nervous system diseases. This is a well written interesting review article, that fits within the scope of the journal. However, the manuscript needs some major revisions prior acceptance for publication in the Cells journal. Please find below the comments/suggestions that will further improve the manuscript:

Specific major comments

-Abstract-Line 19: “highlights the involvement of NKT cells in other cancers and central nervous system diseases”: the involvement of NKT cells in other cancers and CNS diseases is a very general term. Which types? Is there a link of other cancers with GBM? Also, the title refers to GBM only; Please elaborate.  

-Title: if you mention broadly other CNS diseases perhaps you include this term in the title as well.

-Introduction: Lines 58-60: “We will also provide an overview of the role of NKT cells in Multiple Sclerosis, another central nervous system (CNS) disease”: please explain the choice of MS and not other neurological diseases (i.e Parkinson`s or Alzheimer). Is there an association with glioblastoma and a correlation with these diseases or the other cancer types that could be useful in the exploration of the NKT cell mediated mechanisms?

-Table 1: this table is a bit poor in terms of information and a general focus. It would be better to re-arrange and enrich the table with more information. References should have also been included.

-General comment: the role of NKT cells in the hallmarks of cancer should have been emphasized with examples in the NKT cells part.

-Lines 303-305: "On the other hand, paradoxically, in a model of multiple sclerosis, GalCer activated type I NKT cells induce M2 macrophages and MDCSs to ameliorate disease": please elaborate how this is linked with the antitumor immunity. How you can link and extract important information for cancer with the MS knowledge, since they are two different diseases?

-General comment: Type I and II NKT cells sections: These parts could be merged together and you could compare more their function by including a summarized table on the role of NKT in the regulation of tumor immunity (with details on the mechanisms that are implicated and the cancers/GBM that are involved).

-In the NKT Cells in GBM part you mention that these studies are contradictory sometimes; please be more critical in the discussion and analyze the possible reasons and how this may affect their implication in cancer treatments.

-Figure 3: This figure refers to studies involving NKT cell agonists or NKT cells to treat nervous system cancers. Please explain how you would expect similarities or differences in behavior and in terms of immunoregulation, since you have two different types of cancers with different pathophysiological mechanisms.

-General comment: Please explain a bit more in the discussion part (part 7) the role of type I and II NKT cells and how a balance in their mode of action could help the effectiveness in immunotherapies as well as how each type could behave to improve this. Would it be worth to compare other types of cancer with GBM?

Specific minor comments

-Abstract: Line 12: please rephrase as: “Natural Killer T cells (NKT)”

-Introduction: Lines 34-46: Please include some references to further support the statements in the immunotherapeutic part.

-Introduction: Lines 49-53: Please explain a bit more the statement “natural killer T (NKT) cells, which recognize lipid antigens presented by non-classical MHC-like CD1d molecules” and include some references to further support this notion.

-GBM microenvironment:

-Line 129: “These immunosuppressive TAMs”: Please define the immunosuppressive phenotype in GBM and explain the type of macrophages (i.e M0-M1-M2) that may affect the GBM progression.

-Line 137: “Myeloid-derived suppressor cells (MDSCs)”: please mention the most dominant ones, within the GBM TME.

-Line 155-156: “Neutrophils have also been implicated in ferroptosis of GBM tumors, which leads to necrosis of tumor tissue and is associated with poorer survival”: Please explain the mechanism and how the necrosis of tumor tissue can be linked with poorer survival.

- NKT Cells: Lines 240-246: you mention about the single cell RNAseq analysis of human peripheral blood NKT cells suggesting that there may not be such functional subsets in humans. Is there any knowledge over the NKT cells changes within GBM or other cancers tumor microenvironment?

-Lines 388-389: “The two types of NKT cells not only have generally opposing functions in the regulation of tumor immunity, but they also cross-regulate each other”: Please explain more the impact that this may have in terms of cancer progression and future treatments.

Lipids in Brain Immunology and GBM:

-Lines 431-433: “Some lipids, such as gangliosides, are shed from tumor cells and can induce immune responses such as antibody production. Thus, lipids produced by glioma are immunogenic”:  Please explain a bit more here how this is happening and specify the possible link.

-Figure 2 could have been emphasized more within the text.

-General comment over the structure: NKT cells in GBM section should be right after the NKT cells.

- NKT Cells in multiple sclerosis: still if the title refers to GBM, not sure how multiple sclerosis (or other cancers) is linked. Also, what other neurological diseases are affected by NKT cells (see major comments section) and why they have not been mentioned or explored? In general, the help of MS to delineate the GBM mechanisms is not very clear (see major comment).

Author Response

Abstract-Line 19: “highlights the involvement of NKT cells in other cancers and central nervous system diseases”: the involvement of NKT cells in other cancers and CNS diseases is a very general term. Which types? Is there a link of other cancers with GBM? Also, the title refers to GBM only; Please elaborate.

To describe the role of NKT cells in tumor immunity, we discussed knowledge of NKT cells in other types of cancer, as you can read in the NKT cell section.  Thus, we cannot specify the types of cancer, since it is too broad. While we highlight the role of NKT cells in diseases other than glioblastoma in order to provide a broader context for our discussion of NKT cells in glioblastoma, these discussions are supplementary rather than a focal point. The intention of this manuscript is to discuss the role of NKT cells in glioblastoma, as the title indicates.

-Title: if you mention broadly other CNS diseases perhaps you include this term in the title as well.

Since our main focus is on GBM, we do not think it is appropriate to change the title. Including “CNS diseases” or more specifically “MS” in the title would make it necessary to add much more information about the immune landscapes of these broader diseases, which is not the intention of this review.

-Introduction: Lines 58-60: “We will also provide an overview of the role of NKT cells in Multiple Sclerosis, another central nervous system (CNS) disease”: please explain the choice of MS and not other neurological diseases (i.e Parkinson`s or Alzheimer). Is there an association with glioblastoma and a correlation with these diseases or the other cancer types that could be useful in the exploration of the NKT cell mediated mechanisms?

We chose to discuss MS because the pathology is T cell mediated and regulated by NKT cells.  Parkinson’s and Alzheimer are not autoimmune diseases.  We revised the sentence as following:

“As autoimmunity is considered to be the flip side of tumor immunity, we will also provide an overview of the role of NKT cells in Multiple Sclerosis, another central nervous system (CNS) disease mediated by T cells and regulated by NKT cells. Finally, we will consider the primary questions regarding the interactions between NKT cells and GBM that remain unanswered” (Line 63-66).

-Table 1: this table is a bit poor in terms of information and a general focus. It would be better to re-arrange and enrich the table with more information. References should have also been included.

We have expanded the table to clarify that both type I and type II NKT cells are CD1d restricted, as well have added a major mechanism of action by which both exert their effects on cancer. Because this information is considered textbook-level, common knowledge within the NKT field, we do not feel that it is necessary to provide references in this table.

-General comment: the role of NKT cells in the hallmarks of cancer should have been emphasized with examples in the NKT cells part.

We have added the following sentences to clarify this point:

“Evasion of immune surveillance is considered an important hallmark of cancer, and reduced number and function of effector immune cells can contribute to such defective immunological monitoring of tumors (Hanahan and Weinberg, Cell, 2011). In cancer patients, type I NKT cells have been reported to have defects in their number and functions in the peripheral blood” (Line 339-341, and expanded upon in the rest of the paragraph).

-Lines 303-305: "On the other hand, paradoxically, in a model of multiple sclerosis, GalCer activated type I NKT cells induce M2 macrophages and MDCSs to ameliorate disease": please elaborate how this is linked with the antitumor immunity. How you can link and extract important information for cancer with the MS knowledge, since they are two different diseases?

Thank you for pointing this out. In order to avoid unnecessary confusion, we have decided to remove this sentence.

-General comment: Type I and II NKT cells sections: These parts could be merged together and you could compare more their function by including a summarized table on the role of NKT in the regulation of tumor immunity (with details on the mechanisms that are implicated and the cancers/GBM that are involved).

We disagree with this comment and think that discussing type I and type II NKT cells in separate sections creates clarity and organized structure in the manuscript. Furthermore, we synthesize and expand upon the interactions between type I and type II NKT cells in the section entitled “Interactions Between Two Types of NKT Cells and Tumor Immunity” (Line 411).

Regarding the table, we have expanded Table 1 to provide a major mechanism used by each type of NKT cell in regulating cancer.

-In the NKT Cells in GBM part you mention that these studies are contradictory sometimes; please be more critical in the discussion and analyze the possible reasons and how this may affect their implication in cancer treatments.

If the reviewer is referring to the observations on type I NKT cells in human patients, we provided a potential explanation in the following sentences, which we have expanded upon to provide further clarity. We have also emphasized that the role of NKT cells in GBM immunotherapy needs to be further elucidated:

“It should be taken into consideration that the frequency of type I NKT cells in peripheral blood of humans varies over a two-log difference (0.01%-3%) among individuals, and the frequency of type I NKT cells in tissues may show similar amounts of variation. Hence, whether NKT cells can be found in tumor tissues may be a reflection of an individual’s frequency of NKT cells in circulation. However, it may not be critical for NKT cells to infiltrate into tumor tissues, as they can exert anti-tumor functions by producing cytokines to modulate effector T cells that are primed in draining lymph nodes. It remains unknown whether NKT cell levels correlate to response rates to immunotherapies. Thus, the role of NKT cells in natural immunosurveillance and immunotherapy of glioma needs to be further elucidated.” (Lines 519-528).

-Figure 3: This figure refers to studies involving NKT cell agonists or NKT cells to treat nervous system cancers. Please explain how you would expect similarities or differences in behavior and in terms of immunoregulation, since you have two different types of cancers with different pathophysiological mechanisms.

We have further clarified this point in our discussion with the following sentence:

“Altogether, these preclinical studies in various GBM models, as well as neuroblastoma models, suggest that NKT-targeted immunotherapy has the potential to improve the outcome of nervous system cancers (Figure 3)” (Lines 546-548).

-General comment: Please explain a bit more in the discussion part (part 7) the role of type I and II NKT cells and how a balance in their mode of action could help the effectiveness in immunotherapies as well as how each type could behave to improve this. Would it be worth to compare other types of cancer with GBM?

We have added the following sentences to the discussion to further clarify the balance between these therapies:

“Tipping the balance between type I and type II NKT cells to a more type I NKT cell dominant profile, perhaps through the administration of endogenous antigens for type I NKT cells, may provide therapeutic benefit to GBM patients” (Lines 653-656).

“Importantly, single agent treatment with NKT cell-targeting drugs will likely be insufficient to create an immune response against GBM; rather, it will be important to consider combining NKT-targeted therapy with other treatments such as vaccines, chemotherapy, and radiotherapy” (Lines 658-661).

Specific minor comments

-Abstract: Line 12: please rephrase as: “Natural Killer T cells (NKT)”

Revised to Natural Killer T (NKT) cells

-Introduction: Lines 34-46: Please include some references to further support the statements in the immunotherapeutic part.

Thank you for pointing out the missing references here. We have now included them.

-Introduction: Lines 49-53: Please explain a bit more the statement “natural killer T (NKT) cells, which recognize lipid antigens presented by non-classical MHC-like CD1d molecules” and include some references to further support this notion.

We do not understand what additional explanation the reviewer is asking for. Since CD1d is an antigen presenting molecule that present lipids to NKT cells, which the statement means, we cannot provide any more information. We have added references to support this. For further clarity we have revised the sentence to read as follows:

“However, natural killer T (NKT) cells, which recognize lipid antigens presented by non-classical MHC-like CD1d antigen-presenting molecules, have emerged as a significant immune-modulators in tumor immunity” (Lines 53-55).

GBM microenvironment:

-Line 129: “These immunosuppressive TAMs”: Please define the immunosuppressive phenotype in GBM and explain the type of macrophages (i.e M0-M1-M2) that may affect the GBM progression.

The M0-M1-M2 dichotomy is now considered outdated terminology/concept.  Rather than black or white macrophage phenotypes, it is widely accepted within the field that macrophage phenotypes exist on a continuum. We have explained the existence of this continuum within the text:

“The phenotype of these cells can range from pro-tumorigenic to tumor-suppressive, but in GBM, TAMs are commonly skewed toward the immunosuppressive end of the continuum” (Lines 138-140).

-Line 137: “Myeloid-derived suppressor cells (MDSCs)”: please mention the most dominant ones, within the GBM TME.

We have added the following sentence:

Poly-morphonuclear (PMN)-MDSCs were shown to represent about half of CD45+ CD11bhigh cells infiltrating human GBM tumors, though monocytic and neutrophilic MSDCs are also  significantly enriched in human GBM samples” (Lines 148-151).

-Line 155-156: “Neutrophils have also been implicated in ferroptosis of GBM tumors, which leads to necrosis of tumor tissue and is associated with poorer survival”: Please explain the mechanism and how the necrosis of tumor tissue can be linked with poorer survival.

We have further explained the mechanism by adding the following sentence:

“Recently, neutrophils were also implicated in the ferroptosis of GBM tumors, which leads to necrosis of tumor tissue and is associated with poorer survival, though the cause by which necrosis impacts survival is unknown [55]. Neutrophils were shown to transfer granules containing myeloperoxidase into GBM cells, inducing cytotoxicity through the iron-dependent buildup of lipid peroxides” (Lines 168-173).  

- NKT Cells: Lines 240-246: you mention about the single cell RNAseq analysis of human peripheral blood NKT cells suggesting that there may not be such functional subsets in humans. Is there any knowledge over the NKT cells changes within GBM or other cancers tumor microenvironment?

No, observations regarding changed in functional subsets of NKT cells have not been made in cancer patients. Furthermore, we do not discuss NKT cells in cancer patients in this section of the manuscript. However, other changes in NKT cells were described in the section on NKT Cells in Tumor Immunity (Beginning Line 267).  

-Lines 388-389: “The two types of NKT cells not only have generally opposing functions in the regulation of tumor immunity, but they also cross-regulate each other”: Please explain more the impact that this may have in terms of cancer progression and future treatments.

We have already explained this both in this section, as well as in the discussion, in which we have made edits as described above to highlight that the balance between the two types of NKT cells may be important for designing NKT cell-targeted therapies.

Lipids in Brain Immunology and GBM:

-Lines 431-433: “Some lipids, such as gangliosides, are shed from tumor cells and can induce immune responses such as antibody production. Thus, lipids produced by glioma are immunogenic”: Please explain a bit more here how this is happening and specify the possible link.

The fact that patients had antibodies against those lipids means that at least B cell immune responses occurred against those lipids.  Thus, one can conclude that those lipids are immunogenic, which term means that a substance can induce immune response.

We have also added the following sentence to provide further clarification:

“Some lipids, such as gangliosides, are shed from tumor cells [143] and can induce immune responses such as antibody production [144].  The antibody production is presumably elicited by NKTFH providing help to B cells [84,85]” (Lines 463-466).

-Figure 2 could have been emphasized more within the text.

We have now cited the figure multiple times within the text.

-General comment over the structure: NKT cells in GBM section should be right after the NKT cells.

We disagree with the reviewer’s suggestion, as all of the information included before the “NKT cells in GBM” section, including “NKT cells in Tumor Immunology” and “Lipids in Brain Immunology and GBM” sections, provides the contextual information and buildup to allow the reader to better understand what we described in “NKT cells in GBM” section.

- NKT Cells in multiple sclerosis: still if the title refers to GBM, not sure how multiple sclerosis (or other cancers) is linked. Also, what other neurological diseases are affected by NKT cells (see major comments section) and why they have not been mentioned or explored? In general, the help of MS to delineate the GBM mechanisms is not very clear (see major comment).

We provided the rationale above.

Reviewer 2 Report

Brettschneider and Terabe provide a sufficiently detailed overview of the current status of NKT cells in the context of glioma.  They discuss immune landscape of GBM, existence of resident memory CD8+ T cells (TRM) cells within brain parenchyma, presence of different immunosuppressive cells in the GBM tumor microenvironment (TME) such as TAMs, MDSCs and neutrophils. They also discuss T cell activation and suppression by GBM cells and emphasize that GBM cells have altered lipid metabolism and aberrant expression of various lipid products.

Then they, quite logically, discuss NKT cells as CD1d-restricted T cells that recognize lipids as antigens as opposed to peptides.  This could be the most important aspect of the NKT cells, especially in the context of glioma since glioma has a modest reservoir of these cells.  They discuss different types of NKT cells and their potential relevance to GBM.  They provide a succinct account of the mechanism, at both cellular and molecular level, of how NKT cells may be exerting their role, especially in the context of GBM.

They coverage of the pertinent literature is very reasonable.

Finally, they provide a meaningful conclusion and provide pertinent direction/speculation to future research in the area.

I find the manuscript very well written, logical and timely.

Author Response

We thank the reviewer for their positive feedback and enthusiasm for the manuscript.